# Technical, Regulatory, Economic, and Trust Issues Preventing Successful Integration of Sensors into the Mainstream Consumer Wearables Market

**DOI:** 10.3390/s22072731

**Published:** 2022-04-02

**Authors:** Jaime K. Devine, Lindsay P. Schwartz, Steven R. Hursh

**Affiliations:** 1Institutes for Behavior Resources, Inc., Baltimore, MD 21218, USA; lpschwartz@ibrinc.org (L.P.S.); shursh@ibrinc.org (S.R.H.); 2Department of Psychiatry and Behavioral Sciences, Johns Hopkins University School of Medicine, Baltimore, MD 21205, USA

**Keywords:** wearables, economics, technology transfer, mHealth, remote patient monitoring, validation testing

## Abstract

Sensors that track physiological biomarkers of health must be successfully incorporated into a fieldable, wearable device if they are to revolutionize the management of remote patient care and preventative medicine. This perspective article discusses logistical considerations that may impede the process of adapting a body-worn laboratory sensor into a commercial-integrated health monitoring system with a focus on examples from sleep tracking technology.

## 1. Introduction

Wearables stand to revolutionize modern medicine by enabling widespread remote patient monitoring and individualized tools for preventative mobile health (mHealth) care [1,2,3,4]. The pathway for technology transfer seems straight-forward. Recent innovations incorporate sensors that measure health-relevant physiological signals into devices that can easily be worn during everyday life. Algorithms can automatically identify signal features that are known to be health relevant and mobile applications can alert the user or their physician about the user’s health status. Widescale adoption of wearables allows for the accumulation of big data that can then be used to analyze population health statistics or train more advanced computer models for preventive health care. Data can be aggregated from multiple sensors and applications per user and across users to provide an even more robust snapshot of overall health [1,2,5,6,7].

This pathway augurs an efficient automated healthcare system that can reduce the burden of care, increase access to care, and improve overall human quality of life. Additionally, remote patient monitoring is especially desirable in the face of the COVID-19 pandemic. The Sensors journal is full of articles describing scientific, engineering, or computational efforts to improve technology transfer along this pathway, but it is also important to understand practical barriers that may prevent technological advances from successfully integrating with the status quo [3]. The goal of this perspective is to outline logistical obstacles that can arise between the development of sensor technology and the widespread adoption of health-relevant wearable devices in order to better prepare sensor developers to achieve their end goals.

## 2. Sensors vs. Wearables

The term “sensor” can refer to any device that measures and records a physical property, including physiological signals produced by the human body. The term “wearable” has popularly come to refer to body-worn, internet-enabled electronic devices that monitor activity and physiological signals and provide feedback about personal wellbeing to the individual consumer. The most recent generation of wearables process inputs from multiple sensors to create a holistic picture of health.

Just because a multi-sensor wearable measures the same physiological signals as a laboratory counterpart does not mean that those measurements are equivalent. Take, for example, the difference between measuring sleep in a laboratory compared to mobile sleep trackers. The gold standard of sleep measurement, polysomnography (PSG), is a system of sensors used to determine sleep in controlled bedroom environments. Electrodes are attached, usually by glue, to the wearer’s (1) scalp to provide electroencephalography (EEG) readings; (2) face to collect electromyography (EMG) readings from the facial muscles and electro-occulogram (EOG) eye movement readings; (3) chest for electrocardiogram (ECG) readings; and (4) legs for additional EMG muscle readings. PSG also includes chest straps, a nasal canula, and/or a pulse oximeter to monitor breathing rate, respiratory rate, and oxygen saturation. Traditionally, these sensors record data via a wired connection to a central computer and are monitored continuously by a specially trained technician [8,9].

Between the glue, wires, and constant monitoring, it is no surprise that sleep tracking wearables measure sleep using a different set of sensors than PSG. Many modern wrist-worn wearables use accelerometry in combination with photoplethysmography (PPG) to monitor activity and sleep [10,11,12]. PPG detects changes in blood volume based on light absorption by biological tissue [13], in contrast to the electrical signals used in PSG. PPG can provide a measure of multiple physiological signals, effectively replacing the chest straps, nasal canula, pulse oximeter, and ECG commonly used in laboratory PSG recordings while fitting neatly into a wearable device.

PSG and PPG output readings overlap with regard to the physiological signal being measured but the signals are measured using a completely different methodology. In direct comparison against PSG, multi-sensor wearables using PPG fall short when determining sleep stages or identifying periods of wake during a sleep episode [10,14]. Multi-sensor sleep trackers still represent a step forward, but are not accurate enough to be considered replacements for laboratory measurements [10,15]. This is just one example of how laboratory sensors and consumer wearables may differ with regard to measuring signals of interest. Many of the examples outlined in this perspective will focus on sleep tracking use cases. The reason for this is four-fold. Firstly, sleep measurement through wearables is one of the Institute for Behavior Resources (IBR) Operational Fatigue and Performance group’s area of expertise. Secondly, sleep–wake determination requires input from multiple body-worn sensors, as described above, and thus, represents a complex computational case. Thirdly, sleep is related to a litany of health effects. Not only are sleep disturbances correlated to underlying physical and mental health problems, but poor sleep behavior can actually increase the risk of developing a subsequent health issue [16,17,18,19,20]. Because of its relationship to health outcomes and also because sleep behavior can be easily changed by the individual, sleep is an attractive target for mHealth behavioral interventions [21,22,23]. Fourthly, body-worn sensors can more accurately measure a physiological signal if the wearer is in a resting state such as sleep [24,25,26,27]. That is to say, even if the goal is to measure a physiological signal that is unrelated to sleep, that signal may be easier to monitor when the individual is in a quiescent state because there will be less noise from motion artifacts or responses to external stimuli.

In support of this fourth point, a recent review of wearables designed to measure respiratory activity had a relative error percentage in the neighborhood of 10% when measurements were taken during quasi-static conditions such as sitting or sleeping but an error rate closer to 40% during dynamic tasks such as walking [24]. Similarly, in an evaluation of PPG sensors, the absolute error of heart rate (HR) measurement was 30% higher during activity than during rest periods [26]. Sleep episodes are likely to be the longest and most reliable periods of quiescence in humans under real-world conditions; therefore, a wearable system that aims to accurately detect a basal physiological rhythm should consider the benefit of first determining whether the wearer is asleep prior to initiating measurements. Algorithms that determine sleep onset using only wrist activity data have been used for sleep research since the 1990s [28,29,30]. Implementing an activity-based algorithm for sleep–wake determination prior to collecting a signal that is susceptible to motion artifacts may help increase measurement accuracy.

## 3. Wearables vs. Medical Devices

Because most wearables are marketed as consumer accessories rather than as medical devices, device inaccuracy may not necessarily affect market availability. Most wearables are considered general wellness devices rather than medical devices that require governmental approval. Medical devices are similarly defined by the United States Food and Drug Administration (FDA), European Commission Medical Device Regulation (MDR), and International Medical Device Regulators Forum (IMDRF) as being devices intended for the purpose of diagnosis, prevention, monitoring, treatment, prediction, or alleviation of a specific disease or disease state [31,32,33]. Wellness devices, in contrast, encourage a healthy lifestyle without specific relation to disease prevention or treatment. Because they present a low risk to the safety of users, they do not need to be validated against laboratory measurements or approved by a government regulatory body [33]. There appears to be a trend for wearable manufacturers to assert medical applications of their wearables, but those claims will expose the manufacturer to the burden of obtaining certification from the governing body in each jurisdiction in which it is sold.

However, manufacturers of wearables that are not marketed as medical devices may still wish to obtain other forms of certification. For example, the Institute of Electrical and Electronics Engineers (IEEE) has published a draft standard, P360—IEEE Draft Standard for Wearable Consumer Electronic Devices that will focus on “an overview, terminology, and categorization for Wearable Consumer Electronic Devices (or Wearables in short). It further outlines an architecture for a series of standard specifications that define technical requirements and testing methods for different aspects of Wearables, from basic security and suitableness of wear to various functional areas like health, fitness and infotainment etc.” [34].

The breadth of this effort is unclear, but it points to a prospect for the future with independent agencies offering to review and certify devices for specific purposes. While the IEEE standard may focus on the technical engineering aspects of wearables, other groups may focus on the functional attributes of wearables. For example, two joint sleep societies—Sleep Research Society (SRS) and the American Academy of Sleep Medicine (AASM)—regularly discuss the state of sleep tracking technology and publish guidance on acceptability for clinical or research use [11,35,36,37]. Likewise, the International Federation of Sports Medicine (FIMS) has created a quality assurance standard for the application of wearables for physical fitness and athletics [38].

These examples suggest that professional societies have already taken the initiative to self-impose standards for acceptable use of wearables within their own fields. These initiatives can help guide the development and use of wearables at individual steps across the mHealth pipeline but have no regulatory authority over device manufacturers. The application of society standards at the market level will depend on whether manufacturers believe that adopting the standards will be economically beneficial from a marketing perspective.

Whether a wearable is classified as a medical device or a wellness device, medical or professional society endorsement or certification may translate to greater market demand for the product. Conversely, certification, and the process of validation testing to achieve that certification, would likely impose an added cost to wearable production. The manufacturer will have to assume that certification confers sufficient added value to warrant the added cost. Currently, the value of scientific endorsement of a wearable has not been quantified. To that end, IBR has initiated a study to establish the economic value of such validation and endorsement as it pertains to sleep tracking devices [39,40].

## 4. Validation

Many wearables undergo validation testing against laboratory measurements but there are currently no solid criteria for how well a device must perform in order to be considered “valid” [11,41,42]. For sleep tracking technology, wearable systems are usually compared against overnight PSG as well as research-grade actigraphy [10,11], but this form of comparison is not helpful for testing whether a wearable can reliably determine sleep onset for short sleep bouts (i.e., naps) or for unanticipated sleep onset, such as dozing off. An important next step in validation testing against PSG will be to develop a laboratory protocol that replicates erratic sleep schedules. A protocol suggested by researchers at the Walter Reed Army Institute of Research (WRAIR) is to test fieldable sleep trackers against PSG using a modified maintenance of wakefulness test (MWT) [43]. Participants in this protocol would be confined to the laboratory bedroom environment over the course of the study and monitored by PSG even during the day. Any sleep episodes that occurred, including naps, could then be scored by a PSG technician, and compared against the wearable’s ability to detect sleep onset or offset.

There is even less guidance on how to test the validity of wearables outside a controlled environment. This is an important limitation since wearables are designed specifically for use outside the laboratory. Establishing ecological validity requires fit-for-purpose study design and testing within the target population [44]. For sensors that are designed to identify an activity state, such as sleep or exercise, devices can be compared against the wearer’s self-report. Validation testing for IBR’s purpose-built sleep tracker not only included a comparison against the gold-standard PSG, but a subsequent comparison of all-day sleep measurement against self-reported sleep behavior in a population of long-haul pilots [12,45]. Self-report is commonly used to collect data about individuals in the real world, but is not always reliable [44,45,46,47]. Comparing a sensor or wearable’s ability to estimate activity states in real-world environments against self-report is better than no testing at all. It is also worthwhile to note that the user’s perception of accuracy may influence their adoption of wearable technology.

Establishing validity for biological signals that cannot be self-reported, such as HR, requires fit-for-purpose study design and testing within the target population [44]. For example, to test the accuracy of PPG HR sensors across a spectrum of activities and skin tones, the Department of Biomedical Engineering at Duke University recruited participants from diverse racial backgrounds, and gauged skin tone using the Fitzpatrick skin tone scale [26,48]. These participants then wore four different PPG consumer devices and two research-grade ECG sensors during periods of rest, deep breathing, walking, and typing activities. All devices had previously been tested for accuracy at rest, and were time synchronized during the study procedures and analysis. This study is a good example of a protocol designed to test accuracy under a specific use case.

Many recent validation testing studies have simultaneously compared multiple consumer wearables against a laboratory standard at once [10,25,26]. This technique allows multiple devices to be validated using the same study cohort, which is cost-effective, but also allows for a head-to-head comparison between consumer wearables. Inter-device reliability, that is, a comparison of accuracy between two consumer wearables rather than between a wearable and research standard, may be an effective field validation protocol in the future. Considering that wearables are designed for use in the real-world environment, testing for consistency between devices that have already demonstrated measurement accuracy in a controlled setting could be a viable next step for validation.

Developing blanket criteria for ecological validation testing is a daunting and impractical task given the wide applicability of wearable technology for health-monitoring purposes. Instead, guidelines for what constitutes acceptable validity need to be developed independently with respect to specific use cases. Wearables and their applications should be tested for measurement accuracy and efficacy of feedback interventions within the context of their intended real-world usage. Validation testing should be considered a necessary first step not only when manufacturing a new device but also prior to working with data extracted from currently available wearables.

## 5. Manufacturing

Several years ago, IBR partnered with a small company to produce a wearable device that would be suitable for monitoring sleep and fatigue in shiftwork and similar operational work environments [12]. Unfortunately, the company’s other wearable initiatives failed and they ceased manufacturing. This left IBR without the means of producing more devices. The cost of re-creating the design for another device with a new manufacturer was cost-prohibitive and would be even greater in order to update the design with any additional features.

There is tremendous risk in manufacturing a “purpose-built” wearable for a specialty application. From the perspective of the manufacturers, adding new algorithms or sensors to an existing system is undesirable unless the addition promises a sizeable boost in sales and revenue. Specialty applications that are not perceived to represent the interests of the mass market have little chance of increasing revenue. Therefore, even sensor technology which has been shown to be scientifically valid and relevant to health outcomes may not make it to mass production. This consideration is particularly important for mHealth initiatives that depend on widescale user adoption or population-level analysis of health data.

Manufacturing may be carried out at a facility that is in overseas and/or in a different country from the device developer or target consumer population. This distance limits the ability of the developer to provide oversight and quality control in manufacturing. Should the manufacturing of devices be successful, there are still logistical challenges in distribution. The cost of shipping must be factored into the overall price and logistics of manufacturing [49]. This includes determining the method (air or sea) and route for shipping. Quality control during shipping should also be considered. Many manufacturers assume a certain percentage of damaged goods in a shipment to cover the costs of unusable merchandise. The final destination of the devices can provide additional hurdles. Import duties and taxes are levied at varying rates depending on the country of import. Import regulations are different across countries as well: the devices may be classified differently (i.e., medical device vs. wellness device) by legislature in the destination country than from the country of manufacturing, which would affect how their importation is regulated. For example, the definition of a medical device is slightly different in the European Union’s (EU) Medical Device Regulation 2017/745 than in the U.S Food and Drug Cosmetic Act [50]. These logistical challenges are not insurmountable, but require a large capital investment that may not be cost-effective for small companies.

## 6. Working with Existing Data

The high costs and logistical difficulties of manufacturing a purpose-built wearable drives many developers to work with existing wearable data systems. Researchers must then contend with how data are collected and accessed. There is no standard for how current commercial wearables process and display sensor-collected data. One wearable may sample data continuously on an epoch-by-epoch (EBE) basis while another may only sample data at intermittent intervals. Researchers must verify that the frequency of data sampling is sufficient for their analyses.

Data can then either be processed on the device itself or transmitted via wireless signals to a company-owned server or associated mobile application for processing. Most wearable devices do not possess enough processing and battery power to analyze data on the device hardware. The use of cloud-based servers solves this problem by allowing calculations to be carried out on better-powered servers but requires wireless connectivity. A benefit to this approach is that data are stored on cloud-based servers and can be accessed remotely using a company-provided Application Programming Interface (API). APIs allow third-party applications to access data, integrate data from multiple sources, and use their own proprietary algorithms to analyze data.

Using third-party data is not without its pitfalls. Differences between systems may make it difficult to analyze data consistently across devices. Each wearable company has its own API that may provide data as raw EBE activity or as scored summary data. Raw data may not be equivalent between devices due to sampling rate, device sensitivity, or other differences in design. Scored data carry an additional layer of possible variance since non-equivalent raw data are then analyzed using algorithms that may use a different, usually proprietary, set of computations.

The proprietary nature of wearables companies and their algorithms can complicate the use of wearable data for STEMM initiatives. Not only are proprietary algorithms private, but the algorithms and API documentation may be updated by the company without notice. This lack of transparency means that researchers cannot be sure whether data were collected or processed in an accurate or consistent manner and remains a concern for research groups [10].

## 7. Economic Considerations

When designing sensors for wearables, it is worth considering economics in the context of how wearables are built, manufactured, and marketed. The first economic consideration has already been touched upon, namely, manufacturing costs. A sensor that relies on exotic materials or technology will be less successful compared to a sensor that is based on existing and inexpensive technology. After determining the cost of a sensor of interest and its impact on the overall cost of the wearable, the next two factors are (1) perceived market for the sensor information and (2) the perceived impact of the sensor on the market price. In other words, “does the addition of this sensor and information it provides justify a higher price?” Understanding the interplay between marketability of wearables technology and economic demand is an ongoing research initiative at IBR [39,40]. Our initial work assessed the impact of wearable features on demand and indicated that additional features may confer a competitive advantage compared to wearables without that feature despite no change in pricing [51]. What this means is that if the price cannot be increased to cover the cost of an additional sensor, the added cost has to be absorbed by reducing the cost of other components. This dictates, to some extent, the pace at which new sensors will be implemented; new sensors will only be adopted at the rate the cost of existing components is decreased through economies of scale.

Even if the cost of the sensor can be absorbed, there is a cost to redesigning and manufacturing a new version of the device that implements those sensors. That kind of initiative is often driven by the marketing department—who will want the information provided by the sensor, how large is that market, and how much additional revenue will be driven by appealing to that market. Marketing departments have a variety of tools to assess the market value for wearables and new features. Often these market research tools will use panels of consumers to test market the wearable, the sensors, and the new information that it can provide [52,53]. Responses will be skewed by the composition of the panel. For example, most wearable manufacturers see the value of including sleep tracking into a wearable, but consumer panels are likely to be composed of consumers who do not work shifts and thus, have consolidated night-time sleep patterns. Such market testing would not detect the value of measuring daytime sleep behaviors, such as napping, which is a common practice for shift workers and in 24-h operations [45,54,55,56,57]. In this use case, a population that would greatly benefit from sleep tracking (shift workers) may be less likely to adhere to an mHealth sleep hygiene initiative if the devices do not measure sleep duration across the 24-h day.

The ultimate consequence of economic considerations is that it is very difficult to sell a specialty sensor that is designed for a narrow market. Given the economic constraints, it is much more feasible to deliver specialty information from existing sensors using innovative software algorithms, either built into the firmware of the wearable or incorporated into connected applications that can take the raw signal from a general-purpose sensor and extract specialty information. The firmware approach is controlled by the manufacturer and will be limited by their marketing perspective. Connected applications can be used to extract information for specialty markets, but connected applications are limited by the nature of the APIs that manufacturers use to share their sensor information. Specialty applications are forced to derive information from highly condensed and processed information available from the manufacturer’s server. Hopefully, this aspect will change as the emphasis shifts more to health and wellness monitoring and manufacturers see the value in having innovators take their sensor data and apply artificial intelligence to derive important new health metrics. Working collaboratively has huge economic advantages for both the wearable manufacturers and the innovators; it represents a relatively small cost to share data that can drive up demand, and it incentivizes innovators to favor wearables that have open access to raw data.

## 8. Trust

Finally, trust is a significant barrier to the successful integration of wearables into mHealth initiatives. Proprietary algorithms serve as a red flag for human subject research groups, but the protection of proprietary technology is to be expected. Most software programs, including commonly used research tools, have proprietary algorithms and are permitted to update the system ab libitum. A common concern among research groups is that they will not know when the algorithm has been changed in a way that could affect the validity or consistency of their data collection. For example, if a researcher was using a wearable to track menstrual cycle in order to identify the ovulation window in their participants, but the wearable’s algorithm was changed from assuming a 28-day cycle to assuming a 30-day cycle, the results would no longer be accurate. The research team would have no way of knowing that the difference was due to an update since the algorithms are not freely available. The team could risk publishing inaccurate data or drawing false conclusions. Interestingly, while this is often cited as an area of concern when using wearable technology, it is rarely discussed as a concern when using statistical analysis software or other proprietary software systems commonly used in research that could influence the interpretation of results. Similarly, inter-rater reliability of PSG scoring remains an issue even in highly trained technicians [58], which begs the question of the intrinsic accuracy of any interpretation of physiological signal measurements.

Trust in proprietary algorithms may hinge on general distrust for accurate measurement of real-world data. Because researchers cannot control or observe the environment, there is a blind spot in the data collection process. Fostering trust in data science analysis techniques requires clear quantification and communication between science and medical research fields and corporate data entities [59,60]. It also requires trusting individual consumers to use wearables consistently and properly. User adoption and adherence are common challenges for wearable companies and researchers hoping to improve mHealth care pathways [61,62].

Users also need to trust that the feedback provided to them from a wearable is salient to their end goals. Using sleep tracking as an example, wearables that offer an estimation of sleep depth may be appealing because it sounds scientific. Users assume that knowing how many minutes of light, deep, or rapid eye movement (REM) sleep they received will be related to health or medical outcomes. However, preliminary results from IBR’s survey of sleep medicine professionals [40] indicate that the ability of a device to record short naps during the day or night is more desirable to sleep researchers than wearable sleep scoring, which is not the same as clinical PSG sleep staging [10,14]. Despite this, most commercial sleep trackers do not automatically record short sleep episodes, an output that could be relevant to sleep health. It is possible that general consumers are more likely to purchase devices that feature scientifically endorsed features, but the value of scientific endorsement of wearables has not yet been quantified. This is an on-going project at IBR. Public trust in scientists could affect price as well as user adherence if wearables and mHealth platforms commit to promoting scientific validity and health relevance in their products.

Finally, user trust is also dependent on the privacy policies of wearables manufacturers. Third party use of data is typically bound by country-dependent privacy laws. For example, the European Union’s (EU) wide-ranging General Data Protection Regulation laws protect users’ privacy and personal information from any company worldwide that collects data from people residing in the EU [63]. In the United States, a mix of separate laws such as the Health Insurance Portability and Accountability Act (HIPAA) and the Electronic Communications Privacy Act (ECPA) are designed to protect privacy. Furthermore, while many wearable devices are not regulated as medical devices in the United States, they are still capable of collecting health data from individuals. Companies providing a means of collecting wearable health data should, at a bare minimum, communicate with their users by notifying them of how they use their data, if they share or sell data with third parties, and obtain general consent from their users. However, reliance on these practices has been criticized as inadequate and future legislation may enact policies aimed to better protect the user’s data [64]. Privacy protections may look different depending on the user’s geographical location, but they provide a basic level of integrity that can foster trust between the user and the company.

## 9. Conclusions

The goal of this perspective has been to arm researchers and developers with information that will help them overcome logistical hurdles that may prevent the adoption of health-relevant technology into the mainstream. Applications that utilize currently existing devices and/or are agnostic to data formatting may have a better chance of integration on the consumer market. To overcome issues of trust regarding the validity of consumer wearables, applications should not only be tested against laboratory measures, but also demonstrate efficacy in real-world use cases.

The direction of wearables technology development is likely to be driven by marketability concerns on the part of manufacturers despite the significant interest that scientists, clinicians, and technical engineers have in advancing the field. It is therefore important to establish the value of scientific validity to manufacturers who have the means of surmounting the limitations discussed within this perspective. The IBR team’s multi-step project aims to quantify the value of scientifically relevant wearables features in terms of economic value [39,40]. In the future, these findings will be used to help foster collaboration between mHealth researchers and device manufacturers in order to improve the state of the art for wearables technology.

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
