# Peer review of "Technical, Regulatory, Economic, and Trust Issues Preventing Successful Integration of Sensors into the Mainstream Consumer Wearables Market"

_sensors, 2022, doi:10.3390/s22072731_

Round 1

Reviewer 1 Report

The manuscript has been significantly improved. The authors responded to all reviewer´s comments. I have no additional questions or comments.

Author Response

The authors would like to thank the reviewer for their patience and dedication to the improvement of this article. Thank you and have a great day.

Reviewer 2 Report

The paper is a discussion on the different issues with customer wearables, with focus on sleep tracking applications.
As someone who is working in the wearable field I find the discussion relatively interesting, although not surprising or especially surprising.
If this type of article is suitable for this issue of the journal, I have no objection against publishing it, although releasing it as a whitepaper or a blog post could also be appropriate and could help to attract a wider audience.

Minor comments:

    - you should mention the focus on the sleep applications in the abstract
    - "Fourthly, body-worn sensors can more accurately measure a physiological signal if the wearer is in a resting state like sleep" - I believe the key aspect here is in the resting state, which can be detected with a cheap and simple accelerometery; sleep detection is not necessary
    - "Proprietary algorithms serve as a red flag for human subjects’ research groups" - not sure what you mean here, can you give an example?

Author Response

We would like to thank the reviewer for their helpful feedback and time. We had not previously considered the wider potential reach from white paper or blog publication. Thank you for bringing this point to our attention; we will consider this route in the future. Responses to the reviewer's specific comments are outlined below. Have a great day. 

  Comment 1) You should mention the focus on the sleep applications in the abstract

            Response: We have added "with a focus on examples from sleep tracking technology" to the abstract, as per the reviewer's suggestion.

Comment 2) "Fourthly, body-worn sensors can more accurately measure a physiological signal if the wearer is in a resting state like sleep" - I believe the key aspect here is in the resting state, which can be detected with a cheap and simple accelerometery; sleep detection is not necessary

          Response:  The key aspect to this point is that physiological signals are easier to measure when the individual is at rest. Therefore, even if the signal is unrelated to sleep, measurement accuracy may be better if it is measured during sleep or a sedentary period. Our in-text citations reference support from the literature on assessing heart rate and tidal volume while minimizing motion artifacts. We have added a point to this paragraph to clarify our meaning "Fourthly, body-worn sensors can more accurately measure a physiological signal if the wearer is in a resting state like sleep. That is to say, even if the goal is to measure a physiological signal that is unrelated to sleep, that signal may be easier to monitor when the individual is in a quiescent state because there will be less noise from motion artifacts or responses to external stimuli."

Comment  3) "Proprietary algorithms serve as a red flag for human subjects’ research groups" - not sure what you mean here, can you give an example?

        Response: An example has been provided in this section: "A common concern among research groups is that they will not know when the algorithm has been changed in a way that could affect the validity or consistency of their data collection. For example, if a researcher was using a wearable to track menstrual cycle in order to identify the ovulation window in their participants, but the wearable’s algorithm was changed from assuming a 28-day cycle to assuming a 30-day cycle, the results would no longer be accurate. The research team would have no way of knowing that the difference was due to an update since the algorithms are not freely-available. The team could risk publishing inaccurate data or drawing false conclusions.  Interestingly, while this is often cited as an area of concern when using wearable technology, it is rarely discussed as a concern when using statistical analysis software or other proprietary software systems commonly used in research that could influence the interpretation of results."  We hope that this example helps to explain the concerns that researchers may have about proprietary algorithms but also the oversight attached to these concerns.